# Salivary concentrations of secretory leukocyte protease inhibitor and matrix metallopeptidase-9 following a single bout of exercise are associated with intensity and hydration status

**Karen Knipping** [1,2]ᵒ*, **Shirley W. Kartaram**[3]ᵒ, **Marc Teunis**[3], **Nicolaas P. A. Zuithoff**[4], **Nicole Buurman**[1], **Laura M'Rabet**[3], **Klaske van Norren**[5], **Renger Witkamp**[5], **Raymond Pieters**[3,6], **Johan Garssen**[1,2]

1 Danone Nutricia Research, Utrecht, The Netherlands, 2 Department of Pharmaceutical Sciences, Utrecht University, Utrecht, The Netherlands, 3 Research Group Innovative Testing in Life Sciences and Chemistry, University of Applied Sciences Utrecht, Utrecht, The Netherlands, 4 Julius Center for Health Sciences and Primary Care, University Medical Center Utrecht, Utrecht University, Utrecht, The Netherlands, 5 Nutritional Biology, Division Human Nutrition and Health, Wageningen University & Research, Wageningen, The Netherlands, 6 Institute for Risk Assessment Sciences, Immunotoxicology (IRAS), Utrecht University, Utrecht, The Netherlands

ᵒ These authors contributed equally to this work.
* C.T.Knipping@uu.nl

**Data Availability Statement:** Data and the complete statistical analysis as well as all the

## Abstract

### Aim

To investigate the effects of exercise on salivary concentrations of inflammatory markers by analyzing a panel of 25 inflammatory markers in subjects who had participated in bicycle ergometer tests varying in workload and hydration status.

### Methods

Fifteen healthy young men (20–35 years) had performed 4 different exercise protocols of 1 hour duration in a randomly assigned cross-over design, preceded by a rest protocol. Individual workloads depended on participant's pre-assessed individual maximum workload (Wmax): rest (protocol 1), 70% Wmax in hydrated (protocol 2) and dehydrated (protocol 3) state, 50% Wmax (protocol 4) and intermittent 85%/55% Wmax in 2 min blocks (protocol 5). Saliva samples were collected before (T0) and immediately after exercise (T1), and at several time points after exercise (2 hours (T3), 3 hours (T4), 6 hours (T5) and 24 hours (T6)). Secretory Leukocyte Protease Inhibitor (SLPI), Matrix Metallopeptidase-9 (MMP-9) and lactoferrin was analyzed using a commercial ELISA kit, a panel of 22 cytokines and chemokines were analyzed using a commercial multiplex immunoassay. Data was analyzed using a multilevel mixed linear model, with multiple test correction.

figures and tables are available as an R-package and can be installed or downloaded from https://github.com/uashogeschoolutrecht/kinetics/.

**Funding:** This study has been funded by Nationaal Regieorgaan Praktijkgericht Onderzoek SIA, RAAK PRO 4-017. The funders had no role in study design, data collection and analysis, decision to publish, or preparation of the manuscript.

**Competing interests:** Karen Knipping, Nicole Buurman and Johan Garssen are employee of Danone Nutricia Research. All other authors declare no conflict of interest. This does not alter our adherence to PLOS ONE policies on sharing data and materials.

## Results

Among a panel of 25 inflammatory markers, SLPI concentrations were significantly elevated immediately after exercise in all protocols compared to rest and higher concentrations reflected the intensity of exercise and hydration status. MMP-9 showed a significant increase in the 70% Wmax dehydrated, 50% Wmax and intermittent protocols.

## Conclusions

Salivary concentrations of SLPI and MMP-9 seem associated with exercise intensity and hydration status and may offer non-invasive biomarkers to study (local) inflammatory responses to different exercise intensities in human studies.

## Introduction

There is growing interest to assess inflammatory markers in response to different types of stress, including exercise-induced stress [1, 2]. Their usefulness not only lies in the direct applicability to monitoring the degree of stress, but they can also be employed to study immunological processes after (severe) physical stress [3]. Exercise has multiple immunological effects, both in the short term reflected by redistribution of immune cells and changes in circulating levels of cytokines [4] and longer term through either beneficial or unfavorable effects on the immune system and ultimately chronic disease [5]. However, non-exercise factors such as impaired sleep quality, poor nutrition and pollution also cause short-term and prolonged immunological effects, making athletes vulnerable to infections, especially of the upper respiratory tract [6–8]. In case there is a significant disbalance between the effort at the one hand, and rest, fitness and immunological resilience on the other hand, a sustained pathophysiological perturbation, usually called overtraining may arise [9]. Overtraining is a multiple-organ syndrome in which abundance of proinflammatory stimuli lead to a situation where chronic immune suppression and an elevated inflammatory state coincide [4]. Taken together, the complex and sometimes thin lines between beneficial and potentially negative immunological consequences of exercise merit further study.

Exercise-induced immune responses are dependent on different factors including exercise intensity, duration, and the individual's fitness nutrition and hydration status [10–12]. Typically exercise-induced immune-responses include an increase in circulating leukocytes, mainly neutrophils, and increased plasma concentrations of various pro- and anti-inflammatory mediators [13], possibly anti-inflammatory responses are involved in counteracting immune activation, controlling effects of tissue damage and maintaining body function during exercise [14]. Pro-inflammatory mediators includes tumor necrosis factor (TNF)-α, macrophage inflammatory protein (MIP)-1 and interleukin (IL)-1β, and anti-inflammatory cytokines such as IL-6, IL-10, and IL-1-receptor antagonist [5]. In addition, the number of natural killer cells is reduced significantly after exercise and total cytotoxic activity is attenuated [15, 16]. Salivary immunoglobulin A (IgA) secretion rate, a marker of mucosal immunity, is decreased as well [17]. These effects are assumed to have their origin in different organs and tissues, including the respiratory tract, the musculoskeletal apparatus and the intestinal tract. Regarding the latter, we and others have shown that exercise can temporarily cause dysregulation of tight junctions (TJ) and compromise gut barrier integrity [18–20] thereby challenging the immune system which may, consequently, induce inflammatory responses that can become systemic [21]. This exercise-induced dysregulation can be caused by several factors including changes in

blood flow and oxygen delivery to the gut causing ischemia and heat stress causing physical damage to the gut and phosphorylation of the TJ proteins, disturbing their functioning [22–26].

To improve our understanding of the immune/inflammatory events taking place during exercise, saliva could be an easy non-invasive and stress-free tool to evaluate inflammatory markers. It is increasingly acknowledged that saliva can provide a useful matrix for diagnosis of several diseases [27], to study biomarkers of general health [28], and stress [29]. Biomolecules in saliva can have different origins, either via diffusion or active transport from the blood, or locally produced by the salivary glands [30]. Many biomolecules that are circulating in the blood are also found in saliva, sometimes even at higher concentrations than found in blood [31]. Saliva consists of approximately 2,000 different proteins, of which 26% are also found in blood. In terms of infection susceptibility following exercise, saliva may play a role in the innate response, as also reflected by changes in IgA levels [32]. At the same time, the reliability and validity of salivary inflammatory markers in response to stress compared to those determined in blood are still unclear and needs further study [1].

Secretory leukocyte protease inhibitor (SLPI) is a highly abundant protein in mucous secretions in the oral cavity, intestine and respiratory tract. It is part of the superfamily of proteinase inhibitors which cleave peptide bonds. SLPI has been shown to inhibit neutrophil elastase, trypsin, chymase and chymotrypsin; all enzymes that usually cleave peptide bonds important for digestion [33]. Currently, it is also recognized for its anti-inflammatory and antimicrobial role by respectively inhibiting the response of macrophages and monocytes, antagonizing effects of lipopolysaccharides and the interaction with bacterial lipids allowing SLPI to disrupt microbial membranes [34–36]. To our knowledge, SLPI has not been assessed in human exercise studies, but for its anti-inflammatory action and high abundance in saliva it was added to the anti-inflammatory panel.

Matrix metallopeptidase 9 (MMP-9), also known as gelatinase B, is stored in intracellular secretory granules and secreted by several immune cells like neutrophils [37]. The release of MMP-9 from the granules is induced by stimuli such as cytokines (e.g. IL-1$\beta$ and TNF-$\alpha$) [38] and altered cell-matrix contacts [37]. The main effects of MMP-9 are the release of antimicrobial peptides (AMPs) and cytokines, and cleavage of chemokines which can influence chemotaxis and inflammatory reactions [39, 40]. Mahmood et al. analyzed MMP-9 in saliva and plasma at rest and after acute physical exercise in patients with coronary artery disease and found higher levels in saliva compared to plasma but no differences between rest and after exercise [41]. However, a systematic review on MMPs in exercise showed early release of MMP-9 as a result of acute exercise [37]. We therefore argue that MMP-9 is an interesting salivary marker to add to the inflammatory panel.

To our knowledge there is no study that has investigated salivary concentrations of various inflammatory markers following exercise stress at different intensities and hydration status in one and the same study. It is of interest to investigate whether saliva is suitable to identify exercise-induced inflammatory biomarkers to facilitate the applicability of the bicycle ergometer test. It would ease multiple sampling in a short time frame and make the test less demanding for the individuals. Therefore, in the present study the relationship between exercise-stress at different levels of intensity and concentrations of several inflammatory markers, among which SLPI, MMP-9 and lactoferrin, in saliva was investigated.

## Materials and methods

The current study was approved by the medical ethics committee of Wageningen University Research Centre (WUR), The Netherlands, ISRCTN code 13656034, and was conducted according to the Declaration of Helsinki (Fortaleza, Brazil, 2013). Informed consent was signed by the participants. Parts of this study have been published previously [19, 23].

**Table 1. Baseline characteristics and performance data of the 15 subjects.**

| | |
|---|---|
| Age (years) | 24.3 ± 2.4 |
| BMI (kg/m$^2$) | 22.5 ± 1,5 |
| Weight (kg) | 75.8 ± 6.7 |
| Length (cm) | 183.4 ± 3.8 |
| VO$_{2max}$ (ml/kg/min) | 56.9 ± 3.9 |
| W$_{max}$ (W) | 335.1 ± 39.9 |

## Subjects

Fifteen healthy recreational-active male cyclists were selected for this study. The inclusion criteria included non-smoking, the age of 20–35 years, a BMI of between 20 and 25 kg/m2 and with at least 2 years recreational cycling experience. An overview of subjects characteristics are shown in Table 1. They were recruited by means of flyers distributed at the campus of Wageningen University and regional cycling clubs, by word of mouth and via social media. Exclusion criteria were smoking, allergies, gastro-intestinal and immune diseases, use of Controlled Substances and participation in other clinical studies. The subjects were instructed not to perform intense physical activity and not to consume alcohol, two days prior to the test days. To standardize food-intake during the test period, dinners were provided for the evenings before the test days and on the test days. In addition, subjects were requested to keep a diary with training and dietary and illness logs during the whole study period. Dietary habits, including fluid intake, were discussed every test day. Based on a health questionnaire, with earlier mentioned inclusion- and exclusion criteria, subjects were selected for an incremental exercise test. The maximal workload (Wmax) was determined using an electronically braked cycle ergometer (Lode Excalibur, Groningen, The Netherlands). After a short warming-up, the subjects started cycling at 100W with a pedal frequency of 90–100 rotations per minute (RPM). The power increased every minute with 20W until the subject was not able to maintain the workload and pedal frequency dropped to less than 70 RPM.

## Study design

The study was set up in order to investigate the effects of various exercise intensities on the responses of several inflammatory markers. To that end, exercise protocols of 1 hour varying in low- and high-intensity, 50% Wmax, 70% Wmax and 85%/55% Wmax intermittent exercise in blocks of 2 minutes, were randomly assigned in a cross-over design. All subjects started with a rest baseline protocol following the same testing schedule as for the exercise protocols. The high intensity protocol of 70% Wmax was performed in both hydrated and dehydrated condition and both were conducted sequentially in the randomization scheme. The window between the experimental protocols was at least one week. To attain a mild dehydrated condition prior to exercise, subjects were asked to restrict their fluid intake to 0.5 L on the day before the particular test day. Further dehydration was achieved as fluid loss was not compensated during exercise. During exercise in hydrated condition fluid loss was compensated with 3 mL tap water/kg body weight (BW) every 15 min. In the post exercise as well as in the post rest period in the rest condition, the participants drank 200 mL of tap water every hour.

## Test schedule, saliva and blood collection

In the morning, subjects arrived at the laboratory after an overnight fast and events were assessed in the following order. Subjects were asked to sit and relax for approximately 10–15 minutes before obtaining the first sample (T0) which was in fasted condition. Thereafter their

body weight was measured to control weight loss and during exercise. After a light standardized breakfast subjects started, aside from the rest condition as first experimental protocol, with one of the assigned cycling protocols. To ensure mild dehydration, subjects were asked to abstain from any food high in water content [42] and fluid intake was restricted to a maximum intake of 0.5 L fluid on the day prior to the particular test day. Further dehydration was achieved during cycling as the participants did not drink water throughout this particular test. Directly after 1 hour of testing (rest or cycling) body weight was measured to determine post-exercise rehydration corresponding to 150% of body mass loss during exercise. During the remainder of the test day subjects consumed 200 mL of tap water every hour [43]. An extensive lunch was offered after the last blood sampling 5 h post-exercise, after which the participants left.

Saliva samples were collected before (T0) and immediately after exercise (T1), and at several time points after exercise (2 hours (T3), 3 hours (T4), 6 hours (T5) and 24 hours (T6)). Saliva was collected using a swab method; a stick with a sponge-like end (Oracol saliva collection device (oral test kits, S10); Malvern Medical Developments Ltd) was used to absorb saliva into the sponge. The sponge-like end was placed in the mouth between gum and cheek, 5 minutes at the right side followed by 5 minutes at the left side. The saliva collected from both sides was pooled and centrifuged for 5 minutes at 24,000g to remove cells and other non-dissolved components. The supernatant was transferred into a EDTA-tube containing freeze-dried protease inhibitor (Protease inhibitor cocktail Complete Ultra tablets mini; Roche) and mixed well. The saliva was stored at -80˚C until further analysis. Blood samples were collected before, during (0.5 hour) and immediately after (1 hour) of cycling, and at several time points after cycling (1.5 hour, 2 hours, 3 hours, 6 hours, 24 hours) and plasma was stored at -80˚C for analyses of SLPI and MMP-9. Subjects arrived the next morning again fasted at the laboratory for saliva and blood collection of 24 hours to analyze recovery.

## Cytokines, chemokines and growth factors in saliva

Twenty-two cytokines, chemokines and growth factors (IL-1ra, IL-1beta, IL-2, IL-4, IL-5, IL-6, IL-7, IL-8, IL-10, IL-12(p70), IL-13, IL-17a, granulocyte-colony stimulating factor (G-CSF), granulocyte-macrophage colony-stimulating factor (GM-CSF), interferon (IFN)-gamma, tumor necrosis factor (TNF)-alpha, macrophage inflammatory protein (MIP)-1alpha, MIP-1beta, monocyte chemo-attractant protein (MCP)-1, fibroblast growth factors (FGF), platelet-derived growth factor (PDGF)-BB and vascular endothelial growth factor (VEGF)) were determined using a multiplex immunoassay (customized Bio-Plex® Multiplex Immunoassay System, BioRad). This assay was validated for saliva according the EMA guidelines [44]. Saliva samples were measured undiluted in duplicate.

## Matrix metallopeptidase 9 (MMP-9) analysis in saliva and serum

MMP-9 was determined in saliva and serum using a single-plex immunoassay (Human MMP-9 Luminex Performance Assay, R&D Systems) according to the manufacturer's instructions. This assay was validated for saliva according the EMEA guidelines [44]. Saliva sample were prediluted 40x in assay buffer and serum samples were prediluted 50x in assay buffer and measured in duplicate.

## Secretory leukocyte protease inhibitor (SLPI) ELISA in saliva and serum

SLPI was measured using an ELISA for recombinant human SLPI protein (Human SLPI Quantikine ELISA Kit DP100, R&D systems) and performed according to the manufacturer's

protocol. This assay was validated for saliva according the EMA guidelines [44]. Saliva samples were diluted 1500x and serum samples were diluted 20x and measured in duplicate.

### Lactoferrin ELISA using saliva samples

Lactoferrin was measured using the Lactoferrin human in vitro ELISA kit (Human Lactoferrin ELISA Kit (ab200015), Abcam). This assay was validated for saliva according the EMA guidelines [44]. Saliva samples were diluted 1000x and 1500x and measured in duplicate.

### Statistical analysis

Data was analyzed using a linear mixed model (details below). The analysis models the effects of overall protocol differences, differences between the time points within a protocol and the protocol by time interaction. The analyses were performed using the software environment for statistical computing and graphics R [45]. For modelling the R package nlme was used [46]. Model details: Log transformed outcomes were analyzed with a linear mixed model for continuous outcomes. The mixed model was specifically chosen to incorporate repeated measures of the same outcomes over time both during a single protocol and during the entire study (i.e. crossing over to another protocol). We included a random intercept representing differences between participants and a second random intercept representing differences within participants at the start of each new protocol. For some outcomes, the second random intercept was nearly 0 and subsequently removed from the model. As there were clear indications that the progression of the (log transformed) outcomes over time is not linear, time was included in the model as categorical variable. Protocol was also included as a categorical variable; as protocols are expected to influence the outcomes during the follow-up measurements over time, we also included a protocol by time interaction. Distributional assumptions (i.e. normality and homoscedasticity) was assessed with residual analysis [47]. Likelihood ratio (LR) tests were used to determine p-values. These tests were calculated by taking the difference in -2 log likelihood from models with and without the protocol and protocol by time interactions. P-values from LR tests were calculated for each outcome separately. To correct for multiple testing, we applied a false discovery rate (FDR) correction on these p-values [48], under the assumption of possible dependency between p-values. Adjustment was performed only for the primary comparisons (difference between the control and test protocol). Before applying the multiple testing correction, we removed those markers that showed more than 25% missing values to prevent overcorrecting our data.

## Results

### SLPI and MMP-9 in saliva and serum

Salivary concentrations of SLPI showed a significant increase directly at the end of exercise (T1) in all protocols (p = 0.0023) compared to the rest protocol. The largest difference with the rest condition was seen with the dehydrated condition at 70% Wmax. SLPI concentrations were also measured in serum and although concentrations were very low, SLPI in the dehydrated condition remained elevated compared to rest up to 2 hours (T3) post-exercise, which was not seen with the other exercise protocols (Fig 1).

   Concentrations of MMP-9 in saliva showed a significant (p = 0.0136) increase, compared to rest, 1 hour post-exercise (T2) in all protocols but not in the 70% protocol (Fig 2). Compared to saliva, the increase in MMP-9 concentrations in serum was much more pronounced, showing a significant (p<0.0001) increase in all exercise protocols from immediately after exercise

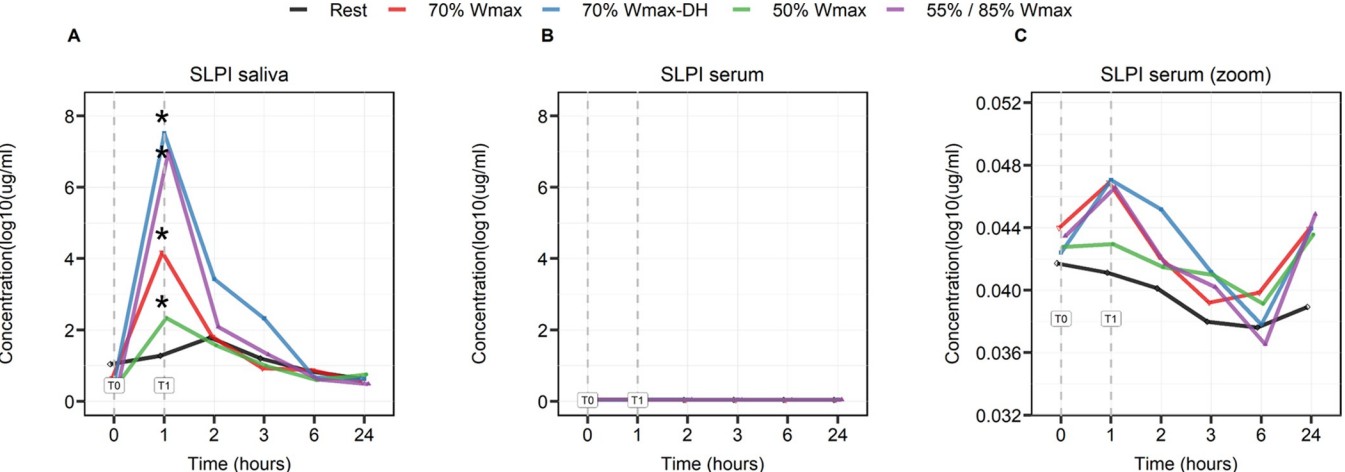

**Fig 1.** Log-transformed SLPI concentrations in saliva (A) and serum (B) and serum adjusted Y-axis (C) in µg/ml before (T0), at the end (1 hour) of cycling (T1), and at several time points after cycling (2, 3, 6 and 24 hours). The effects are shown as mean between the subjects. Significance in SLPI concentrations in saliva at T1 is found in all protocols (* p = 0.0023) compared to the rest protocol after multiple testing.

(T1), peaking 2 hours post-exercise (T3), and slowly declining at 6 hours in all protocols compared to the rest protocol.

## Cytokines/chemokines/growth factors and lactoferrin in saliva

Lactoferrin showed significant (p = 0.0049) increase in concentrations only in dehydrated condition at 70% Wmax immediately after exercise (T1) and post-exercise (T2 and T3) compared to the rest protocol (data not shown), however significance disappeared after correction for multiple testing. IL-6 showed a significant (p = 0.0317) increase in the dehydrated condition at 70% Wmax but only immediately after exercise (T1) (data not shown). However, differences were not significant after FDR correction. None of the other markers showed any significant

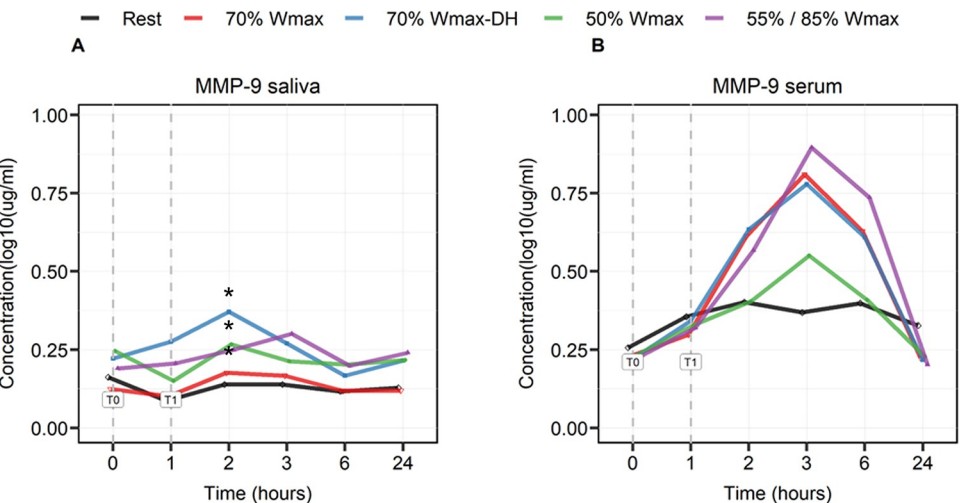

**Fig 2.** Log-transformed MMP-9 concentrations in saliva (A) and serum (B) in µg/ml before (T0), immediately after exercise (T1), and at several time points after exercise (2, 3, 6 and 24 hours). The effects are shown as mean between the subjects. Significance in MMP-9 concentrations in saliva at T2 was found in all protocols (* p = 0.0136), with the exception of 70%, compared to the rest protocol after multiple testing.

**Table 2. Results from likelihood ratio test for determining p-values with (FDR) and without (Raw) correction for multiple testing of the different markers.**

| Saliva marker | Raw | False discovery rate (FDR) |
|---|---|---|
| SLPI | < .0001 | 0.0023 |
| SLPI serum | 0.1726 | 1.0000 |
| MMP-9 | 0.0136 | 0.1975 |
| MMP-9 serum | < .0001 | < .0001 |
| Lactoferrin | 0.0049 | 0.1174 |
| IL-1beta | 0.1204 | 1.0000 |
| IL-2 | 0.4393 | 1.0000 |
| IL-4 | 0.7548 | 1.0000 |
| IL-5 | 0.7873 | 1.0000 |
| IL-6 | 0.0317 | 0.1975 |
| IL-8 | 0.4927 | 1.0000 |
| IL-10 | 0.6299 | 1.0000 |
| IL-12(p70) | 0.4427 | 1.0000 |
| IL-13 | 0.4543 | 1.0000 |
| IL-17a | 0.8049 | 1.0000 |
| G-CSF | 0.1948 | 1.0000 |
| GM-CSF | 0.6965 | 1.0000 |
| IFN-gamma | 0.7676 | 1.0000 |
| TNF-alpha | 0.6687 | 1.0000 |
| MIP-1beta | 0.1292 | 1.0000 |

differences after correction of the p-value for multiple comparisons between the rest protocol and the exercise protocols (Table 2).

## Discussion

Results from our study demonstrate that salivary SLPI and MMP-9 concentrations show significant and differential increases between the different exercise protocols and therefore, offers interesting opportunities to be further investigated as potential biomarker to study exercise-induced effects on the immune system. From a panel of 22 immune-markers, these 2 parameters showed a response that was related to the workload. In case of SLPI the response in saliva was clearly more pronounced than that in serum. Compared to blood, collection of saliva is non-invasive, does not require the presence of a trained professional and may be performed under field conditions. Apart from these practical advantages, analysis of inflammatory markers in saliva can provide specific information, for example on processes taking place in the oral cavity, upper airways and gastro-intestinal tract. At the same time, the reliability and validity of saliva to measure inflammatory markers in response to stress is still unclear and not validated in most cases. An example on how to validate the biological variation and diagnostic accuracy of dehydration assessment markers is described by Cheuvront et al. [49]. Many studies have demonstrated that exercise induces acute and marked immune responses [6, 50, 51]. For the purpose of studying these immune markers, standardized exercise models are attractive to study effects of clinical physical stressors [52].

It is known that exercise can have both positive and negative effects on susceptibility to infection. Acute bouts of exercise cause a temporary depression/increase of various aspects of immune function (e.g., neutrophil respiratory burst, lymphocyte proliferation, monocyte TLR, and major histocompatibility complex class II protein expression) that can last 3–24 hours after exercise, depending on the intensity and duration of the exercise bout [50].

To our knowledge this is the first study that reports the relationship between the different exercise tests varying in workload and hydration status and saliva and plasma levels of SLPI and MMP-9 during and after exercise up to 24 hours. SLPI, which is found in the highest concentrations in saliva, is a protein that exhibits antimicrobial and anti-inflammatory activities to protect local tissue and is thought to play a critical role in mucosal defense [34]. In our study, the higher levels of salivary SLPI immediately after exercise compared to serum, indicate that SLPI might be an early stress marker in saliva. In saliva, a strong increase was seen at the end of the exercise (T1) with the highest levels seen in dehydrated condition at 70% Wmax and after intermittent exercise at 85/55% Wmax. In dehydrated condition at 70% Wmax SLPI levels were still elevated 1–2 hours post-exercise (T2 and T3), whereas in all other exercise protocols the levels returned back to baseline at the timepoint 1 hour post-exercise (T2). So far, only one previous study investigated SLPI in an exercise model using Wistar rats, in which an increase in mRNA SLPI expression was found [53].

MMP-9 is an anti-inflammatory mediator regulating tissue remodeling by directly degrading extracellular matrix proteins, activation of cytokines and chemokines and mediating leukocyte migration during inflammation [54]. In our study, MMP-9 in saliva shows a clear peak at 1 hour after exercise (T2) in the protocols 70% Wmax in dehydrated condition, 50% Wmax and intermittent exercise 85/55% Wmax. MMP-9 levels in serum were also increased 1 hour post-exercise (T2), but in all test situations its levels peaked at 2 hours post-exercise (T3) followed by a slow decline towards 24 hours. Human saliva is considered a plasma ultra-filtrate and contains proteins either synthesized in situ in the salivary glands or derived from blood. Hence, it contains biomarkers derived from serum, gingival crevicular fluid, and mucosal transudate [55]. In our study, MMP-9 levels in saliva do not directly reflect blood concentrations since the pattern of increase is different in saliva from that in plasma. It is known that MMP-9 activity is regulated by different mechanisms [56, 57], including modulation of transcription, activation of pro-MMP-9 and inhibition by tissue inhibitors of MMPs-TIMPs [58]. The current data could indicate that MMP-9 in saliva might be produced (locally) an anti-inflammatory response by mechanisms and cells that differ from those responsible for its release in plasma. Elevated MMP-9 serum levels could be indicative for an increase of intestinal permeability induced by the exercise [59–61]. Previous studies analyzing either plasma or skeletal muscle have shown that the response of MMP-9 to resistance exercise is dependent upon the duration of the training [62] and it was found that a single bout of exercise increases levels of activated MMP-9 in skeletal muscle and in the circulation [54]. Mahmood et al. compared MMP-9 levels in saliva and plasma after acute physical exercise and reported 2- to 4-fold higher levels of MMP-9 in saliva compared to plasma 30 minutes after exercise [41]. Our data however shows the opposite with a 2- to 3-fold higher levels in plasma 1–2 hours after exercise respectively. Overall, the increased salivary and plasma MMP-9 levels may reflect an enhanced potential of the mucosal immune defense.

Salivary lactoferrin and IL-6 were affected by exercise as well, although to a much lesser extent than SLPI and MMP-9. Lactoferrin, an antimicrobial protein, is part of the innate defense mainly at the mucosal level. In our study, lactoferrin showed differences only in dehydrated condition at 70% Wmax at the end of exercise (T1) and post-exercise (T2 and T3). A previous study with acute running showed an increased salivary lactoferrin concentration which might result in a decreased infection susceptibility, adding further evidence for positive effects on the immune system of short-term exercise [63]. IL-6 is an interleukin considered both a pro-inflammatory cytokine and an anti-inflammatory myokine. In our study, IL-6 showed a difference in the dehydrated condition at 70% Wmax only at the end of exercise (T1). Measurement of IL-6 has been widely undertaken to examine inflammatory and immune responses to exercise since physical exercise is associated with elevation of serum levels of IL-6

because of its production in the muscles. Previously it has been shown that IL-6 concentrations in serum/plasma and saliva were not correlated [64, 65], possibly due to the lack of relationships between the systemic/muscular and the salivary routes of IL-6 production. However, both lactoferrin and IL-6 did not show significant differences after FDR corrections, however they could be interesting for further investigation in exercise.

One could suggest that the high salivary levels of SLPI are a result of dehydration which could lead to a more concentrated saliva. Defining (de)hydration is difficult as the function and storage of fluid as total body water is distributed between the intracellular fluid and extracellular fluid compartments. Terms such as "euhydration" "hypohydration" and "dehydration" are typically referring to whole body water content. However, water is stored in many different compartments not only the intracellular, interstitial, and plasma spaces, but also the gastrointestinal tract and bladder and the location of fluid will influence its function. A recent review on assessment of hydration in athletes describes methods that range in validity and reliability, from complicated and invasive methods to moderately invasive blood, urine and salivary variables. Their conclusion was that any single assessment of hydration status is problematic and the recommended approach is to use a combination to increase accuracy and validity [66]. Blood plasma osmolality, plasma sodium concentration or blood serum osmolality are regularly used blood markers for evaluating hydration status for the differential diagnosis of disorders related to the hydrolytic balance regulation, renal function, and small-molecule poisonings, and the dehydration status in blood of this study was already discussed in a previous publication [19]. Villiger et al. describe that during exercise and heat exposures, saliva might be an effective index to evaluate hydration status but seems to be highly variable and should be carefully used as a substitute marker of other biochemical hydration assessment markers [67]. Taylor et al. concluded that while expectorated saliva osmolality tracked mass losses within individuals, its large intra- and inter-individual variability limited its predictive power and sensitivity, rendering its utility questionable [68]. In our study we used swabs to collect saliva as we collected saliva during the last 5 minutes of exercise. As the flow rate is also an often used parameter in saliva, this was not applicable in our study as we can't estimate whether or when the swab was completely saturated. Several studies have been done to investigate leakage of markers from blood into saliva. For example, Killer et al. investigated the effects of hydration status on saliva antimicrobial peptides in responses to endurance exercise. Their finding was that exercise in dehydrated state caused a reduction in saliva flow rate yet induced greater secretion rates of lactoferrin and higher concentrations of secretory IgA and lysozyme. Dehydration does not impair saliva antimicrobial peptide responses to endurance exercise [69]. Thus, no standardized and validated hydration marker in saliva is available and future research is needed to discover better methods to assess the hydration status in saliva.

The current study was an adaptation of exercise-induced stress in well-trained healthy young men described by JanssenDuijghuijsen [70]. Both studies were primarily focused on the effect of exercise on stress-related markers such as intestinal integrity markers and myokines. JanssenDuijghuisen found that here seems to be an adaptation to exercise-induced stress or damage responses in well-trained healthy young men. This finding may explain, in part, inconsistent outcomes in the literature concerning several factors of exercise-induced stress. It also has implications for the design of protocols to assess exercise-induced responses. The current study adapted the exercise protocols still with a focus on intestinal integrity. The addition of saliva and measurement of inflammatory markers in this study was exploratory. Therefore, the set-up might not be the optimal set-up for saliva. One could speculate that in dehydrated state the saliva could be more concentrated, resulting in higher measured levels of the markers. However, this should then be the case for all measured markers and in our study we only find this specifically for SLPI, and also only in saliva and not in blood. Therefore we hypothesize

that the exercise-induced increase of SLPI which is the highest in the dehydrated protocol might be a valuable salivary marker that needs further exploration.

## Conclusion

SLPI and MMP-9 have potential as salivary markers to monitor relative exercise load under field conditions. The increased concentration of AMPs (SLPI and lactoferrin) in saliva which might result from synergistic compensation within the mucosal immune system may confer some benefit to the non-specific immune response. To validate levels of SLPI and MMP9 in saliva as biomarkers for (local) inflammatory responses in humans, further studies to investigate salivary levels of SLPI and MMP-9 in relation to chronic inflammation (overtraining, burn-out, IBD, obesity, age and infections commonly seen after strenuous exercise, *i.e.* upper respiratory tract infections) are needed.

## Acknowledgments

We would like to thank all subjects for their commitment during the study. Furthermore, we would like to thank Anne Geijssen for her excellent support in the collection and technical analysis of the samples.

## Author Contributions

**Conceptualization:** Karen Knipping, Shirley W. Kartaram, Laura M'Rabet, Klaske van Norren, Renger Witkamp, Raymond Pieters, Johan Garssen.

**Data curation:** Marc Teunis, Nicolaas P. A. Zuithoff.

**Formal analysis:** Nicolaas P. A. Zuithoff, Nicole Buurman.

**Funding acquisition:** Johan Garssen.

**Investigation:** Karen Knipping, Shirley W. Kartaram.

**Methodology:** Shirley W. Kartaram, Nicole Buurman.

**Software:** Marc Teunis.

**Supervision:** Renger Witkamp, Raymond Pieters, Johan Garssen.

**Validation:** Karen Knipping.

**Visualization:** Marc Teunis.

**Writing – original draft:** Karen Knipping.

**Writing – review & editing:** Shirley W. Kartaram, Marc Teunis, Nicolaas P. A. Zuithoff, Laura M'Rabet, Klaske van Norren, Renger Witkamp, Raymond Pieters, Johan Garssen.

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
