## [Decision Letter · Decision Letter 0]

21 Jul 2022

PONE-D-22-17078Salivary levels of Secretory Leukocyte Protease Inhibitor and Matrix Metallopeptidase-9 following a single bout of exercise are associated with intensity and hydration statusPLOS ONE

Dear Dr. Knipping,

Thank you for submitting your manuscript to PLOS ONE. After careful consideration, we feel that it has merit but does not fully meet PLOS ONE’s publication criteria as it currently stands. Therefore, we invite you to submit a revised version of the manuscript that addresses the points raised during the review process.

ACADEMIC EDITOR: This paper is well written and the results would be a beneficial contribution to the body of knowledge. Please address the reviewers comments in your resubmission. 

We look forward to receiving your revised manuscript.

Kind regards,

William M. Adams

Academic Editor

PLOS ONE

Journal Requirements:

a) Did participants provide their written or verbal informed consent to participate in this study?

"This study has been funded by The Dutch Society of Sciences and Art NWO SIA, RAAK project RAAK PRO 4-017"

5. Thank you for stating the following in the Conflict of interest and funding Section of your manuscript: 

"Karen Knipping, Nicole Buurman and Johan Garssen are employee of Danone Nutricia Research. All other authors declare no conflict of interest. This study has been funded by The Dutch Society of Sciences and Art NWO SIA, RAAK project RAAK PRO 4-017"

"This study has been funded by The Dutch Society of Sciences and Art NWO SIA, RAAK project RAAK PRO 4-017"

6. Thank you for stating the following in the Competing Interests section: 

"Karen Knipping, Nicole Buurman and Johan Garssen are employee of Danone Nutricia Research. All other authors declare no conflict of interest."

Reviewers' comments:

Reviewer's Responses to Questions

**Comments to the Author**

1. Is the manuscript technically sound, and do the data support the conclusions?

Reviewer #1: Partly

Reviewer #2: Yes

2. Has the statistical analysis been performed appropriately and rigorously? 

Reviewer #1: Yes

Reviewer #2: Yes

3. Have the authors made all data underlying the findings in their manuscript fully available?

Reviewer #1: No

Reviewer #2: Yes

4. Is the manuscript presented in an intelligible fashion and written in standard English?

Reviewer #1: Yes

Reviewer #2: Yes

5. Review Comments to the Author

Reviewer #1: Thanks for the opportunity to review this work. The authors present a study in which healthy individuals undertook one rest and four exercise protocols to investigate a relatively broad spectrum of immune/inflammatory biomarker responses to four exercise protocols.

Having referred to the other papers arising from this design, it appears the study has been conducted to a high standard and generates novel data that will contribute useful information to the field. However, I do not fully understand the rationale for slicing the work into multiple publications. Much of the important information required to interpret the present findings has already been published (e.g. subject characteristics; Borg RPE; hydration status as well as known immune markers such as immune cell counts). This means that this paper reads with a lot of seemingly missing data that has already been published elsewhere. I understand the authors were probably striving to avoid double publication of results, but repetition of certain details in my opinion is essential in order provide the full picture for the reader. For example, the glutamine bolus administered in Kartaram (2019) is omitted completely in this paper – could this modulate immune biomarkers and thus be of importance to the reader? In fact, I would say the findings here complement the findings of the paper of Kartaram (2020) well and it is a shame to see them separated out. Sometimes the value of the sum of the parts is lesser than the whole.

However, it is my role as a reviewer to give recommendations to the authors as to how this specific manuscript could be improved. I do believe the data present novel value and that the manuscript has potential for publication. Nevertheless I have a relatively long list further comments that I hope the authors find constructive and will consider should they choose to revise the manuscript. Please refer to the attached file for detailed comments.

Reviewer #2: This manuscript mainly provides an investigation of salivary and serum inflammatory markers concentrations in response to strenuous exercise varying in intensity and hydration status. To this end, fifteen healthy young men were subjected to five different exercise protocols of one hour duration in a randomly assigned cross-over design separated by at least 1 week of wash out. Fluid restriction prior, -during and post-exercise was applied in the dehydrated condition. Data was analyzed using a multilevel mixed linear model and corrected to account for multiple testing. Among the inflammatory markers investigated, SLPI and MMP-9 showed some associations with the intensity of exercise and hydration status, during and post exercise. In particular, the authors report that saliva SLPI levels were significantly elevated immediately after exercise in all protocols compared to rest. Values reflected the intensity of exercise and hydration status. Additionally, serum MMP-9 showed a significant increase in the 70% Wmax dehydrated condition, 50% Wmax and intermittent protocols. The authors conclude that

The paper is generally well written, the study design is comprehensive and the methodologies to assess the variation of inflammatory markers well described. However, the article has some questions to be addressed.

Comments

*Methods

Line 125 Hydration status is a known confounder of exercise-induced responses. The authors mention that food-intake was standardized during the test period in the form of standardized meals provided on the evenings before the test days and the days of the test. However it is unclear how fluid intake was controlled prior to the intervention?

Lines 144-145 It is said that hydration status was evaluated the day of the test via a measure of body weight loss. The dehydration protocol also includes fluid restriction the day before the test day, described at lines 140-141. Body weight measurement at baseline is unlikely discriminate underhydrated participants from well hydrated participants. How did the authors objectively check hydration status in the morning of the test day ? This appears to be missing in the manuscript and is an important point considering that hydration status 1. plays a role in exercise-induced responses and 2. is modulated in the design of the present investigation.

Lines 145-146 “a baseline sample of” ? The authors should detail the type of sample they refer to.

*Results

- Line 261 The authors are invited to elaborate on the “significant differences” observed for lactoferrin (increase? decrease?)

*Discussion

Lines 254-255 How do the authors explain the absence of increase in salivary MMP-9 in the 70% condition whereas the increase is obtained for all other test conditions including 50 Wmax? It is suggested that the authors provide some explanations in the discussion.

Line 339 It is suggested to elaborate on this sentence. What is implied by “a result of dehydration”? Do the authors mean a decrease in total body water which directly and mechanically leads to an increased concentration of molecules in body fluids?

Line 340 This sentence is a subjective broad statement. The function and storage of fluid throughout the body is not complicated. Please rephrase.

Lines 348-350 The authors are invited to specify in which conditions these blood parameters are used to assess hydration status (e.g acute changes in total body water). For example, plasma osmolality is not an accurate marker of hydration in the general population where large differences in daily water intake do not translate into differences in plasma osmolality, which is tightly maintained within a narrow range.

Lines 361-364 Please articulate these two sentences for better clarity; suggestions: “thereby showing”, “which shows”…

Lines 369-370 This sentence is unclear and somewhat cumbersome. One cannot hypothesize that something is interesting. The authors are invited to rephrase this sentence

*General flow

Chronological order at lines 144-145 - The sentence “body weight was measured to control weight loss and hydration status during exercise” should be placed after all the sentences describing the events occurring before body weight measurement. Unless body weight is measured first thing in the morning once the subject has arrived at the clinic. But in that case also it should be clarified.

Lines 278 – 285 The authors detail here the pros of performing saliva samples vs blood samples and the relevance in the context of their study. It feels this section would be more appropriate if moved later in the discussion, in a paragraph elaborating on the strengths and weaknesses of the present investigation.

*Dehydrated condition

Dehydrated condition: the description start in the “study design section”, is repeated at lines 148-149. The authors are encouraged to describe the dehydrated condition protocol exhaustively in the same part of the manuscript.

Lines 150-152 Post-exercise hydration instructions are detailed in a way that suggests they apply to all test conditions. Were the subjects following the dehydrated test also rehydrated post-exercise? + Did they have access to water during the exercise?

Minor comments

Line 31 Suggest adding hydration in the keywords

Line 45 Suggest adding Wmax in the abstract “70% Wmax dehydrated”

There are some typos/grammatical errors in the manuscript, so recommend authors to read the manuscript again, e.g. :

- Lines 69-70 Redundance of “in addition” and “also”

- Lines 70-71 Redundance of ‘In addition” and “as well”

- Line 115 “to” is missing

- Line 253: “a more abundant”

- Line 300-301: this sentence is heavy

- Line 336-337 Redundance of “however” and “but”

- Line 350: “the dehydration status in blood parameters” this wording is cumbersome

- Line 361 “there” � “their”?

6. PLOS authors have the option to publish the peer review history of their article (what does this mean?). If published, this will include your full peer review and any attached files.

Reviewer #1: **Yes: **Helen G Hanstock PhD

Reviewer #2: No

---

## [Author Response · Author response to Decision Letter 0]

14 Apr 2023

Reviewer #1

General comments:

Thanks for the opportunity to review this work. The authors present a study in which healthy individuals undertook one rest and four exercise protocols to investigate a relatively broad spectrum of immune/inflammatory biomarker responses to four exercise protocols.

Having referred to the other papers arising from this design, it appears the study has been conducted to a high standard and generates useful data that will contribute novel observations to the field. However, I do not fully understand the rationale for slicing the work into multiple publications. Much of the important information required to interpret the present findings has already been published (e.g. subject characteristics; Borg RPE; hydration status as well as known immune markers such as immune cell counts). This means that this paper reads with a lot of seemingly missing data that has already been published. I understand the authors were probably striving to avoid double publication of results, but repetition of certain details in my opinion is essential in order provide the full picture for the reader. For example, the glutamine bolus administered in Kartaram (2019) is omitted completely in this paper – could this modulate immune biomarkers and thus be of importance to the reader? In fact, I would say the findings here complement the findings of the paper of Kartaram (2020) well and it is a shame to see them separated out. Sometimes the value of the sum of the parts is lesser than the whole.

However, it is my role as a reviewer to give recommendations to the authors as to how this specific manuscript could be improved. I do believe the data present novel value and that the manuscript has potential for publication. Nevertheless I have a relatively long list further comments that I hope the authors find constructive and will consider should they choose to revise the manuscript.

Twenty two inflammatory markers were measured in saliva and/or serum but the whole rationale and abstract focuses on just two – SLPI and MMP-9. Were these main aims established a priori or post-hoc based on the results? Please be clear throughout the manuscript on what were the main (a priori) aims and hypothesis as well as secondary aims and any exploratory or post-hoc analyses. It should help to then relate aspects of e.g. design choices and the weight given to particular results/conclusions back to the aims.

The statistical analyses are described in careful detail but then the results lack transparency with a lack of p values for specific fixed factors and interactions as well as a lack of raw and supporting data.

The rigour of the scientific writing could do with additional review, to ensure precision of terms and arguments and that references are provided where appropriate. In addition, the results must be reported to a much greater degree of specificity. There are also a handful of small grammatical errors and quirks in the text and thus it would benefit from a final review for clarity as well as correctness.

Dear Reviewer #1,

We would like to express our sincere thanks for the extensive review. The reviewer really did his/her best to improve the quality of our manuscript and put a lot of time and energy into this. Unfortunately, such high quality reviews are nowadays more the exception than the rule, with reviewers sometimes taking on the role of gatekeepers rather than peers who do his/her best to help colleagues. We are really grateful to you! We have implemented the various suggestions and have also thoroughly adapted the text once again. Please find our reactions to the different comments below.

Specific comments:

Abstract

L35: See above comment - was the aim to investigate just these two proteins (SLPI and MMP-9) or a whole panel of 22 markers (of which these two came out as significant)? Changed to ‘To investigate the effects of strenuous exercise on salivary levels by analyzing a panel of 22 inflammatory markers in subjects who had participated in bicycle ergometer tests varying in workload and hydration status.’

L38: ”exercise protocols of 1 hour duration in a randomly assigned cross-over design of their individual Wmax” – this reads a bit strangely as the design isn’t related to Wmax – perhaps “based on their individual Wmax” or consider beginning a new sentence to highlight the individual protocols/intensities? Changed to ‘Fifteen healthy young men (20-35 years) had performed 4 different exercise protocols of 1 hour duration in a randomly assigned cross-over design, preceded by a rest protocol. Individual workloads depended on participant’s pre-assessed individual maximum workload (Wmax): rest (protocol 1), 70% Wmax in hydrated (protocol 2) and dehydrated (protocol 3) state, 50% Wmax (protocol 4) and intermittent 85%/55% Wmax in 2 min blocks (protocol 5).’

L40: can you include the specific time-points for saliva sampling? Changed to ‘Saliva samples were collected before (T0) and at the end (1 hour) of cycling (T1), and at several time points after cycling (2 hours (T3), 3 hours (T4), 6 hours (T5) and 24 hours (T6))’

L44: “Values” of what? Please be specific. Changed to ‘and higher concentrations reflected…..’

L46: Conclusions – It is a bit of a jump to go from an association with intensity and hydration status to a biomarker of immunity in general. Please consider carefully the specific conclusions that can be drawn from the observations in the present study. L47: the exercise is described as “strenuous” but is this descriptor appropriate for 1 h at e.g. 50% Wmax? L46 and L47 Changed to ‘Salivary levels of SLPI and MMP-9 seem associated with exercise intensity and hydration status and may offer non-invasive biomarkers to study immune responses to different exercise intensities in human studies.’

Introduction

I miss a specific rationale for how SLPI and MMP-9 secretion in saliva would be hypothesized to be affected by exercise (added to the dedicated sections of SLPI and MMP-9 in the introduction), given what is known about other immune responses to exercise. What makes you think that these markers might be associated with exercise load – or are they simply important aspects of functional immunity that haven’t been previously measured?

L51 – Please include a reference for the first sentence if possible. 2 References on salivary inflammatory markers in (exercise-induced) stress are added

L52 – I am curious as to any examples of existing biomarkers that can be provided here that reliably and accurately monitor “stress load”. It appears to me that although the field has been looking for such markers for some time, none (especially not in saliva) have really achieved this yet – at least not for individual monitoring and decision-making. The reviewer is right that there are no specific biomarkers existing for stress load. We have changed it to ‘the degree of stress’.

L54 – Reference 1 (Pedersen et al) is a relatively old study that I am not sure accurately supports the statement made in this sentence. Reference replaced with: ‘The Immunomodulatory Effects of Physical Activity’. Kruger et al. Curr Pharm Des. 2016;22(24):3730-48.

L54-55 – “Exercise has multiple effects on the immune system” – is a quite vague statement and I wonder if you can start by being any more specific? Changed to ‘Exercise has multiple immunological effects, both in the short term reflected by redistribution of immune cells and changes in circulating levels of cytokines [4] and longer term through either beneficial or unfavorable effects on the immune system and ultimately chronic disease [5].’

L56 – “…strenuous exercise can temporarily impair both the innate and the adaptive immune system” – this assumption has been contended in recent years (see e.g. Campbell and Turner 2018 Front Immunol; Simpson et al 2020 Ex Immunol Rev). Once again, it would be helpful to be more specific about the contexts of “strenuous” exercise that have been shown to disturb immune function – exercise can be characterized by several other prescription factors (e.g. intensity, duration, modality, continuous/intermittent) and I suspect “strenuous” may not be sufficiently specific to describe the relationship between exercise and immunity. As the authors note in the following paragraph, exercise-induced responses are dependent on intensity, duration and hydration (among other things – I would contend also modality and intermittent/continuous prescriptions are important factors). Campbell and Turner 2018 Front Immunol; Simpson et al 2020 Ex Immunol Rev argue that stress by exercise does not give immune suppression but probably an redistribution of leucocytes between tissues and circulation. Simpson states that susceptibility for infection is also under the influence of other factors like sleep, circadian rhythm, diet etc. It has been changed to ‘However, non-exercise factors such as sleep quality, poor nutrition and pollution (Simpson et al) can affect the short - and long term effects and might compromise immune function, making athletes vulnerable to infections, especially of the upper respiratory tract’.

L58 – sentence beginning “In case” needs a reference. Added: The proinflammatory effects of chronic excessive exercise. Da Rocha et al. Cytokine. 2019

L61 – “immune distortion” – correct terminology? Changed to ‘Chronic immune suppression’

L61 – “increased systemic inflammatory “state” – can a “state” be increased? ‘State’ changed to ‘response’

L61 – what “important roles”? Changed to ‘Overtraining is a multiple-organ syndrome in which finally an abundance of proinflammatory cytokines cause chronic immune suppression and chronic inflammation [4].

L64 – sentence beginning “Exercise-induced responses” needs a reference as well as the following sentence. In addition, the whole paragraph that follows could benefit from specific reference to these prescription factors (intensity, duration and hydration status) when presenting what is known about exercise-induced immune responses. Several reference s added.

L66 – if the plasma concentration of both pro and anti-inflammatory mediators increases, what does this mean for ‘net’ inflammation? Added ‘suggesting that such responses may be required for counteracting the inflammatory threat, control of tissue damage and maintenance of body function during prolonged strenuous exercise [12]’

L77 – “aspects” � “factors”? Adapted

L80 – “find new markers” – of what, specifically? There might not even be a need for ’new markers’ indeed, sentence changed to ’To improve our understanding of the immune/inflammatory events taking place during exercise, saliva could be an easy non-invasive and stress-free tool to evaluate inflammatory markers.

L81 – “it is increasingly acknowledged that saliva often can provides (Adapted) a useful matrix to study biomarkers of general health, stress and other variables”. This sentence could do with a reference, but I would contend that a) use of salivary biomarkers is not new – for example salivary IgA was measured in 1982 by Tomasi in cross-country skiers and b) saliva as a matrix has also been relatively frequently criticized as highly variable and susceptible to confounding factors, as the authors themselves note in the discussion. Jos Bosch provides an excellent review of some additional confounding mechanisms for salivary biomarkers in his chapter in Saliva Monographs in Oral Science (2014) – available here: 10.1159/000358864. Consideration of the secretory mechanisms of the novel proteins presented here might benefit how their responses to exercise can be interpreted. Changed to ‘It is increasingly acknowledged that saliva can provides a useful matrix for diagnosis of several diseases [25], to study biomarkers of general health [26], and stress [27]. Biomolecules in saliva can have different origins, either via diffusion or active transport from the blood, or locally produced by the salivary glands (refs added). 

L100-105 – It is unclear how the statements about other heterogeneous diseases (CV disease, periodontitis, cancer and IBD) and the further discussion about the role of MMP-9 and IBD is relevant in relation to your hypothesis and research question. The sentence ‘In case of IBD, the increase in serum MMP-9 levels has been suggested to cause degradation of the tract tissue. A possible mechanism of action might be the degradation of occludin which is part of the TJs responsible for a functional gut barrier and integrity’ has been replaced with ‘Mahmood et al. analyzed MMP-9 in saliva and plasma at rest and after acute physical exercise in patients with coronary artery disease and found higher levels in saliva compared to plasma but no differences between rest and after exercise [33]. However, a systematic review on MMPs in exercise showed early release of MMP-9 as a result of acute exercise making it an interesting salivary marker to add to the inflammatory panel.’

L109 – the aim should be stated clearly, ideally accompanied by a hypothesis, that in turn is substantiated with further specific background to inform the research question. Changed to ‘To our knowledge there is no study that has investigated salivary concentrations of various inflammatory markers following exercise stress at different intensities and hydration status in one and the same study. It is of interest to investigate whether saliva is suitable to identify exercise-induced inflammatory biomarkers to facilitate the applicability of the bicycle ergometer test. It would ease multiple sampling in a short time frame and make the test less demanding for the individuals. Therefore, in the present study the relationship between exercise-stress at different levels of intensity and concentrations of several inflammatory markers, among which SLPI, MMP-9 and lactoferrin, in saliva was investigated.’

Methods 

“Subjects and Methods” seems like an unusual header for this section, I would imagine “methods” would suffice unless specified by the journal. Changed to ‘Materials and Methods’

Please report at minimum age, body mass, health status and fitness level of the participants in the present paper. Readers should not have to retrieve this information from previous publications. No clear rationale is provided for the four selected exercise protocols and it would be nice if the protocol selection became more apparent to the reader based on the rationale presented in the introduction. L129 – how were subjects selected? Please detail. To the ‘Subjects’ heading is added throughout the alinea ‘The inclusion criteria included non-smoking, the age of 20-35 years, a BMI of between 20 and 25 kg/m2 and with at least 2 years recreational cycling experience. An overview of subjects characteristics are shown in Table 1.’ ‘Dietary habits, including fluid intake, were discussed every test day.’ ‘Based on a health questionnaire with earlier mentioned inclusion- and exclusion criteria, subjects were selected for an incremental exercise test.’

Please be consistent with the reference to time points. E.g. T1 is referred to as “at the end of cycling” (L163), “directly at the end of exercise” (L248), and elsewhere simply “at the end of exercise”. Using either “at T1” or “immediately after exercise” would probably suffice throughout. Changed to ‘immediately after exercise (T1)’.

L126 – “and at on test days themselves” Adjusted

L133 – “felled back” is not correct English, suggest “and pedal frequency dropped/decreased below 70 RPM”. Changed to ‘and pedal frequency dropped/decreased below 70 RPM’

L136 – Please briefly explain the rationale for the semi-crossover design, was something determined at rested baseline? Additionally, rather than using “protocol 1-5” it might be more intuitive for the reader to follow if you were to describe the exercise protocols by the specific variables that are being manipulated between protocols. Added ‘The study was set up in order to investigate the effects of various exercise intensities on the responses of several inflammatory markers. To that end, exercise protocols of 1 hour varying in low- and high-intensity, 50% Wmax, 70% Wmax and 85%/55% Wmax intermittent exercise in blocks of 2 minutes, were randomly assigned in a cross-over design.’

L144 – was body mass measured nude throughout? Weight was taken in tight underpants

L148 – how was baseline hydration status (pre-exercise) determined after one day of prior fluid restriction? Changed to ‘To ensure mild dehydration, subjects were asked to abstain from any food high in water content [45] and fluid intake was restricted to a maximum intake of 0.5 L fluid on the day prior to the particular test day. Further dehydration was achieved during cycling as the participants did not drink water throughout this particular test’.

L152 – is it possible to provide a reference for 200ml tap water per hour to maintain hydration status? Ref added: Sawka, M. N., Burke, L. M., Eichner, E. R., Maughan, R. J., Montain, S. J., & Stachenfeld, N. S. (2007). Exercise and fluid replacement. In Medicine and Science in Sports and Exercise (Vol. 39, Issue 2, pp. 377–390).

L159 – “collected at” � “collected from”. Adapted

L174-183; L189-196; L200-208; L212-220 – I suggest the assay details could be left out, as they should be available in a kit insert. It would be more helpful to provide information about whether samples were analysed in single, duplicate, triplicate, and provide within and between plate CVs if available. Were all samples in range? I would also suggest to retain the dilution factors for the samples. Please also provide kit IDs and full supplier details where appropriate. We have removed the details descriptions of the assays and added whether they were all measured in duplicate (dilutions were already in) Detailed information about the used kit is added. The kits were validated according the EMA guidelines which contains e.g. intra- and inter-assay variation. If CVs were not according guidelines the kits would not have been used. During validation the dilutions are determined to be mostly within the linear part of the reference curve, samples above detection limit are repeated in a higher dilution, samples below detection limit receive a value of LOD/2.

L242 – please define the “primary comparisons”. Changed to ‘ Adjustment was performed only for the primary comparisons (difference between the control and test protocol).

L244 – can you provide more details about the missing values? Were they randomly or systematically missing or “unquantified knowns” (i.e. samples below assay detection limits)? Samples below detection limit are assigned a number of LOD/2 and are therefore not missing data. There are several types of missing values, not all subjects produced enough saliva at some timepoints to perform all analysis. For the cytokines/chemokines we had the opportunity for the second run of multiplex to add several analytes (FGF, PDGF-BB, VEGF, IL-1ra, MIP-1a) to the multiplex assay which was not possible during the first analysis, meaning that these are missing for some participants (not enough sample to redo analysis with this new multiplex).

Results

The results contain no p values or quantitative statements in-text. This prevents proper scrutiny of the statistical analysis and interpretation of the results. Please consider supporting statements with statistics and other quantitative statements as well as references to appropriate figures and tables. 

No data are provided about the exercise protocols. Did the participants respond as expected? What was measured to ensure the protocols were delivered correctly – have you any measures of e.g. heart rate, metabolic rate (amino acids and urea were analysed as metabolic markers, described in Kartaram et al. Front Physiol 2020;11:1006), substrate utilization, power output, perceived exertion? How did these differ between the protocols? Volunteers’ perceived exertion during exercise was intensity-dependent. After 30 min cycling at high intensity exercise volunteers scored high on the RPE (Rating of Perceived Exertion scale). High intensity exercise in mildly dehydrated condition was perceived to be the most exhaustive. One volunteer did not complete the high intensity exercise protocol at 70%Wmax in dehydrated condition due to personal and practical issues. This volunteer, however, was included because data analysis showed no differences in the means (described in Kartaram et al. Front Physiol 2020;11:1006).

Please also report the mean and variance of achieved dehydration status of the participants in the dehydration exercise trial. From Kartaram et al (2020), it appears that body mass loss during exercise was 1.4% (difference between trials c. 1%) with no notable change in body mass resulting from the pre-exercise fluid restriction. I would argue that this is at the minimum limit to describe a dehydration intervention (1% difference between trials) – and should at best be referred to as mild dehydration as in the previous paper. The authors agree with the reviewer and it is now described as mild dehydration in the paper.

L248 – Salivary “levels” of SLPI – is this a quantified concentration and if so, I suggest you use “concentration” with appropriate units. ‘Levels’ have been replaced with ‘concentrations’

Discussion

The authors discuss the lack of validation of salivary biomarkers. I recommend the authors consider incorporating evaluation of/discussion of biological variability statistics used to evaluate the utility of candidate biomarkers in clinical chemistry for both population cut-off and biological monitoring applications. See Cheuvront et al (2010) Am J Clin Nutr for an example of application of these statistics in a sports science/hydration context. Added: ‘An example on how to validate the biological variation and diagnostic accuracy of dehydration assessment markers is described by Cheuvront et al. [49].’

The authors mention that they could not obtain flow rates, but did the authors consider correcting for e.g. total protein or osmolarity in the saliva samples to account for artefacts created by e.g. changes in saliva flow rate during dehydration or after exercise? In previous studies we measured total protein in saliva with different methods, unfortunately with different results. It should be validated first which method is best suitable for the saliva matrix. Osmolality was considered in a later stage, however, there are too many samples missing to be able to perform calculation per sample.

It would be helpful if the authors could justify the choice to not include any known immune responses to exercise for comparison in the present study. How is one supposed to evaluate the response of novel immune markers to relatively novel protocols without a benchmark series of biomarkers that would be expected to respond based on current understanding? The study was intended to investigate the relationships between exercise intensity and the blood kinetics of different physiological parameters in healthy human volunteers including parameters indicative of metabolic activity (e.g., creatinine, bicarbonate), immunological and hematological functionality (e.g., leukocytes, hemoglobin) and intestinal physiology (citrulline, intestinal fatty acid-binding protein, and zonulin). The addition of saliva was purely exploratory, and although saliva is used for biomarker measurements very frequently, there is no standardized exercise-related saliva biomarker known. SLPI was included in the panel to possibly serve as a reference protein to use for normalization (as albumin is used in serum), as it is always present in high levels in saliva and no literature was known on exercise influence. You can understand our surprise that in this study SLPI showed the biggest differences between protocols. Would it be possible to incorporate previous results from the same study protocol in the discussion, for example? I have had another look at the other parameters to see if we could possibly connect, however the plasma citrulline and iFABP are correlated to intestinal function/damage and is not relevant for this saliva investigation.

Lactoferrin only responded significantly to the dehydrated condition. Could this be an artefact of salivary flow rate leading to an acute concentration effect purely due to fluid losses from saliva secretions? This has been a point of discussion, however, since we don’t see this concentration of markers due to dehydration in any of the other markers the dehydration exercise protocol, we don’t think this is the case.

IL-6 is described as a myokine produced in the muscle and tends to produce a robust response to most types of exercise in plasma/serum. Was IL-6 measured in serum/plasma in the present study? Il-6 was not measured in serum. Given the authors cite a study showing low correlation between serum and saliva responses, and that the IL-6 response in saliva in the present study was relatively low, could the authors evaluate the likely utility of measuring a marker produced by exercising muscle in saliva? In short - could saliva as a matrix too far removed for measurement of exercise-induced IL-6 responses? Indeed it is found by other groups that salivary IL-6 does not reflect blood IL-6 and it seems quite unlikely that salivary IL-6 could be a surrogate marker for blood IL-6, possibly due to the lack of relationships between the systemic/muscular and the salivary routes of IL-6 production (part after , is added to the discussion).

The paragraph on dehydration (from L339) seems relatively far removed from discussion of the findings of the present study in context. At L364 a study is cited that found dehydration did not impair salivary AMP responses to endurance exercise. At L366 the authors speculate that dehydration would drive salivary biomarker concentrations higher. Did you expect that the protocol following fluid restriction would exacerbate biomarker responses and if so, based on a “true” stressor on the immune system or as an artefact of fluid balance shifts? As these saliva markers were exploratory, there was no defined expectation of behavior of these markers following fluid restriction. There might even be other aspects involved in exercise and saliva which has not been studied extensively, perhaps breathing through the mouth increases the exposure of microbes to the mouth mucosal, thereby stimulating AMP’s as a first defense. We found this was too far away from the question of the study to incorporate in the discussion.

The discussion lacks a full evaluation of strengths and weaknesses of the study design. The current study was an adaptation of exercise-induced stress in well-trained healthy young men described by JanssenDuijghuijsen (Exp Physiol. 2017 Jan 1;102(1):86-99). Both studies were primarily focused on the effect of exercise on stress-related markers such as intestinal integrity markers and myokines. JanssenDuijghuisen found that here seems to be an adaptation to exercise-induced stress or damage responses in well-trained healthy young men. This finding may explain, in part, inconsistent outcomes in the literature concerning several factors of exercise-induced stress. It also has implications for the design of protocols to assess exercise-induced responses. The current study adapted the exercise protocols still with a focus on intestinal integrity. The addition of saliva and measurement of inflammatory markers (possibly to be connected to gut health markers) in this study was exploratory. Therefore, the set-up might not be the optimal set-up for saliva, however, interesting findings were nevertheless discovered (added to discussion).

L274 – This statement that SLPI and MMP-9 warrant further investigation as biomarkers of exercise effects on immune function requires further justification. Why is this the case? How do the data enable you to conclude this? Changed to: ’Results from our study demonstrate that salivary SLPI and MMP-9 concentrations show significant and differential increases between the different exercise protocols and therefore offers interesting opportunities to be further investigated as potential biomarker to study exercise-induced effects on the immune system

L277 – It needs to be clearer in the results section how the responses of SLPI and MMP-9 “relate to the workload”. Workload in the protocols was not incremental but involved manipulation of multiple distinct prescription factors such as hydration, intensity and continuous/intermittent exercise. Please consider how this “relationship” could be statistically demonstrated from your data. I have discussed this point with the statisticians who have done the statistical analysis of this study and it is not possible to demonstrate relation to workload unfortunately.

L278 – does blood collection always require a trained professional (e.g. if collecting capillary samples – which can also be done in the field?) If you want to measure many different analytes in the blood, high volumes are needed and therefore trained personal for blood withdrawal would be necessary. If only wanting to measure 1 or 2 specific analytes you could maybe use capillary samples.

L282 – If the authors have 5 repeated baselines measured under controlled conditions, would this give additional useful information when assessing the biomarkers’ suitability for diagnostics/monitoring applications? Almost all markers show comparable T0 levels and levels will also return to ‘normal’ level at T24 with limited or no changes during the in-between timepoints. Exception are a few cytokines, e.g. IL-6 who were already high at T0 in some subjects and returned to lower levels at T24 which could be due to stress/anxiety before starting a endurance protocol. This should however be better investigated by e.g. questionnaires on stress before start of exercise.

L284 – please explain the jump from previously demonstrated immune responses to exercise to the need/attraction of standardised exercise models. Added: ‘For the purpose of studying these immune markers,’

L291 – please consider how your data demonstrate this “relationship” between workload and SLPI and MMP-9 concentrations. Changed ‘workload’ to ‘the different exercise tests varying in workload and hydration status’

L295 – it is unclear how absolute higher concentrations in saliva indicate that SLPI is an “early stress marker”, please explain. Changed to ‘the higher levels of salivary SLPI immediately after exercise compared to serum, indicate that SLPI might be an early stress marker in saliva’

L315 – sentence needs a reference – which cells and mechanisms? 2 refs added

L325 – how do you determine that these markers were affected to a lesser extent? Fair comment, we have removed ‘to a lesser extent’

L357 – flow rate not possible, but correction for osmolarity or total protein would have been? As explained a few comments above: In previous studies we measured total protein in saliva with different methods, unfortunately with different results. It should be validated first which method is best suitable for the saliva matrix. Osmolality was considered in a later stage, however, there are too many samples missing to be able to perform calculation per sample.

L368 – is blood osmolarity more tightly homeostatically regulated than e.g. saliva composition? There is not much known from literature whether blood osmolarity/osmolality is more tightly regulated than saliva (study in spaceflight ongoing), the mechanism is however very different from saliva (water to move from plasma through acinar cells to form primary saliva, a trans-acinar cell sodium gradient must be generated) whereas increased blood osmolality will stimulate antidiuretic hormone for increased water absorption.

L369 – is this statement a hypothesis, a result, or a question for the future? (Thus, no standardized and validated hydration marker in saliva is available and future research is needed to discover better methods to assess the hydration status in saliva.) This is a recommendation for further research.

Conclusion

It is not clear from the discussion how the observations of biomarker responses to exercise in various exercise protocols in the present study lead the authors to conclude that the markers have potential. It could therefore be helpful to define the criteria or expected results for a biomarker that has potential a priori, based on a logical rationale and/or associations with other outcomes, to enable this conclusion later.

L374 – suggest that the applications are either taken in the discussion. The authors are not sure what the reviewers requests here.

L376 – this statement is a nice summary but could do with more mechanistic support from the discussion (The increased concentration of AMPs (SLPI and lactoferrin) in saliva which might result from synergistic compensation within the mucosal immune system may confer some benefit to the non-specific immune response.) Also here we don’t want to overcomplicate the discussion by adding another factor of exposure to microbes via breathing through the mouth.

Tables

The caption of Table 1 displays “model statistics” but actually only raw and adjusted p values are given. It is not clear what the p values in the table represent. If only overall model fit, where are the p values for the main effects and interactions?

Is it possible to add concentration data to Table 1 or another table as appropriate? All data is freely available, the table would become very extensive when adding it for the main article.

Figures

Significantly different pairwise comparisons between protocols and timepoints do not appear to be represented on the figures, which is a shame. Please consider including these graphically or at the very least in the figure captions. Done in figure caption.

Figure 1b appears surplus – it is appropriate to consider that absolute concentrations vary between sample matrices. The zoom is shown that there is a certain variation between de different timepoints and protocols, however not significant, figure B is to show that SLPI is measurable in very high levels when compared to serum.

If it would be possible to provide a y axis with back-transformed raw concentrations on a log scale, this would aid interpretation in terms of normative values for the reader.

Data availability

I tried to access the link provided to the R code/statistical analyses and the link appeared dead. Please check this! The link was still set to ‘private’, but is not set to ‘open’.

Reviewer #2

Reviewer #2: This manuscript mainly provides an investigation of salivary and serum inflammatory markers concentrations in response to strenuous exercise varying in intensity and hydration status. To this end, fifteen healthy young men were subjected to five different exercise protocols of one hour duration in a randomly assigned cross-over design separated by at least 1 week of wash out. Fluid restriction prior, -during and post-exercise was applied in the dehydrated condition. Data was analyzed using a multilevel mixed linear model and corrected to account for multiple testing. Among the inflammatory markers investigated, SLPI and MMP-9 showed some associations with the intensity of exercise and hydration status, during and post exercise. In particular, the authors report that saliva SLPI levels were significantly elevated immediately after exercise in all protocols compared to rest. Values reflected the intensity of exercise and hydration status. Additionally, serum MMP-9 showed a significant increase in the 70% Wmax dehydrated condition, 50% Wmax and intermittent protocols. The authors conclude that

The paper is generally well written, the study design is comprehensive and the methodologies to assess the variation of inflammatory markers well described. However, the article has some questions to be addressed.

Dear Reviewer #2,

We would like to express our sincere thanks for the extensive review. The reviewer really did his/her best to improve the quality of our manuscript and put a lot of time and energy into this. Unfortunately, such high quality reviews are nowadays more the exception than the rule, with reviewers sometimes taking on the role of gatekeepers rather than peers who do his/her best to help colleagues. We are really grateful to you! We have implemented the various suggestions and have also thoroughly adapted the text once again. Please find our reactions to the different comments below.

Comments

*Methods

Line 125 Hydration status is a known confounder of exercise-induced responses. The authors mention that food-intake was standardized during the test period in the form of standardized meals provided on the evenings before the test days and the days of the test. However it is unclear how fluid intake was controlled prior to the intervention? Dietary habits, including fluid intake, were discussed every test day. To achieve a mild dehydrated condition the participants were restricted to a maximum intake of 0.5 L fluid on the day prior to the particular test day.

Lines 144-145 It is said that hydration status was evaluated the day of the test via a measure of body weight loss. The dehydration protocol also includes fluid restriction the day before the test day, described at lines 140-141. Body weight measurement at baseline is unlikely discriminate underhydrated participants from well hydrated participants. How did the authors objectively check hydration status in the morning of the test day? This appears to be missing in the manuscript and is an important point considering that hydration status 1. plays a role in exercise-induced responses and 2. is modulated in the design of the present investigation. Added ‘To ensure mild dehydration, subjects were asked to abstain from any food high in water content (Lopez, 2011) and fluid intake was restricted to a maximum intake of 0.5 L fluid on the day prior to the particular test day. Further dehydration was achieved during cycling as the participants did not drink water throughout this particular test.’

Lines 145-146 “a baseline sample of” ? The authors should detail the type of sample they refer to. Changed to ‘Subjects were asked to sit and relax for approximately 10-15 minutes before obtaining the first sample (T0) which was in fasted condition’.

*Results

- Line 261 The authors are invited to elaborate on the “significant differences” observed for lactoferrin (increase? decrease?). For both Lactoferrin and IL-6 there was a significant increase in concentration in the dehydrated 70% protocol, lactoferrin at different timepoints (T1, T2, T3) and IL-6 only at T1. There differences were lost after FDR correction. To the text is added that there was an increase in concentration.

*Discussion

Lines 254-255 How do the authors explain the absence of increase in salivary MMP-9 in the 70% condition whereas the increase is obtained for all other test conditions including 50 Wmax? It is suggested that the authors provide some explanations in the discussion. When looking at the figure, the 70% and 70% DH does not seem to differ in pattern, however after multiple testing it does not show significance, in which case we cannot state there is a difference.

Line 339 It is suggested to elaborate on this sentence. What is implied by “a result of dehydration”? Do the authors mean a decrease in total body water which directly and mechanically leads to an increased concentration of molecules in body fluids? Changed to ‘’ One could suggest that the high salivary levels of SLPI are a result of dehydration which could lead to a more concentrated saliva’

Line 340 This sentence is a subjective broad statement. The function and storage of fluid throughout the body is not complicated. Please rephrase. ‘throughout the body is complicated’ is rephrased to ‘as total body water is distributed between the intracellular fluid and extracellular fluid compartments.’

Lines 348-350 The authors are invited to specify in which conditions these blood parameters are used to assess hydration status (e.g acute changes in total body water). For example, plasma osmolality is not an accurate marker of hydration in the general population where large differences in daily water intake do not translate into differences in plasma osmolality, which is tightly maintained within a narrow range. Added: ‘for the differential diagnosis of disorders related to the hydrolytic balance regulation, renal function, and small-molecule poisonings,’

Lines 361-364 Please articulate these two sentences for better clarity; suggestions: “thereby showing”, “which shows”…

Lines 369-370 This sentence is unclear and somewhat cumbersome. One cannot hypothesize that something is interesting. The authors are invited to rephrase this sentence. Changed to ‘Therefore we hypothesize that the exercise-induced increase of SLPI which is the highest in the dehydrated protocol might be a valuable salivary marker that needs further exploration’.

*General flow

Chronological order at lines 144-145 - The sentence “body weight was measured to control weight loss and hydration status during exercise” should be placed after all the sentences describing the events occurring before body weight measurement. Unless body weight is measured first thing in the morning once the subject has arrived at the clinic. But in that case also it should be clarified. Events have been described in the order they were performed.

Lines 278 – 285 The authors detail here the pros of performing saliva samples vs blood samples and the relevance in the context of their study. It feels this section would be more appropriate if moved later in the discussion, in a paragraph elaborating on the strengths and weaknesses of the present investigation. The current study was an adaptation of exercise-induced stress in well-trained healthy young men described by JanssenDuijghuijsen (Exp Physiol. 2017 Jan 1;102(1):86-99). Both studies were primarily focused on the effect of exercise on stress-related markers such as intestinal integrity markers and myokines. JanssenDuijghuisen found that here seems to be an adaptation to exercise-induced stress or damage responses in well-trained healthy young men. This finding may explain, in part, inconsistent outcomes in the literature concerning several factors of exercise-induced stress. It also has implications for the design of protocols to assess exercise-induced responses. The current study adapted the exercise protocols still with a focus on intestinal integrity. The addition of saliva and measurement of inflammatory markers (possibly to be connected to gut health markers) in this study was exploratory. Therefore, the set-up might not be the optimal set-up for saliva, however, interesting findings were nevertheless discovered (added to discussion).

*Dehydrated condition

Dehydrated condition: the description start in the “study design section”, is repeated at lines 148-149. The authors are encouraged to describe the dehydrated condition protocol exhaustively in the same part of the manuscript. According the reviewer’s remark, the ‘Study design’ section was rewritten.

Lines 150-152 Post-exercise hydration instructions are detailed in a way that suggests they apply to all test conditions. Were the subjects following the dehydrated test also rehydrated post-exercise? Yes. Did they have access to water during the exercise? Only the 7-% dehydrated protocol had no access to water during exercise, the other protocols did.

Minor comments

Line 31 Suggest adding hydration in the keywords. Added

Line 45 Suggest adding Wmax in the abstract “70% Wmax dehydrated”. Added

There are some typos/grammatical errors in the manuscript, so recommend authors to read the manuscript again, e.g. :

- Lines 69-70 Redundance of “in addition” and “also”. Adapted

- Lines 70-71 Redundance of ‘In addition” and “as well”. Adapted

- Line 115 “to” is missing. Added

- Line 253: “a more abundant” Changed to ‘show higher concentrations in’

- Line 300-301: this sentence is heavy

- Line 336-337 Redundance of “however” and “but”. Adapted

- Line 350: “the dehydration status in blood parameters” this wording is cumbersome. The sentence had been changes, also according comment of another reviewer.

- Line 361 “there” � “their”? Adapted

---

## [Decision Letter · Decision Letter 1]

14 Jun 2023

PONE-D-22-17078R1Salivary concentrations of Secretory Leukocyte Protease Inhibitor and Matrix Metallopeptidase-9 following a single bout of exercise are associated with intensity and hydration statusPLOS ONE

Dear Dr. Knipping,

Thank you for submitting your manuscript to PLOS ONE. After careful consideration, we feel that it has merit but does not fully meet PLOS ONE’s publication criteria as it currently stands. Therefore, we invite you to submit a revised version of the manuscript that addresses the points raised during the review process.

ACADEMIC EDITOR: Please insert comments here and delete this placeholder text when finished. Be sure to:Please address the comments from the reviewers==============================

We look forward to receiving your revised manuscript.

Kind regards,

William M. Adams

Academic Editor

PLOS ONE

Journal Requirements:

Reviewers' comments:

Reviewer's Responses to Questions

**Comments to the Author**

1. If the authors have adequately addressed your comments raised in a previous round of review and you feel that this manuscript is now acceptable for publication, you may indicate that here to bypass the “Comments to the Author” section, enter your conflict of interest statement in the “Confidential to Editor” section, and submit your "Accept" recommendation.

Reviewer #3: All comments have been addressed

Reviewer #4: All comments have been addressed

2. Is the manuscript technically sound, and do the data support the conclusions?

Reviewer #3: Yes

Reviewer #4: Yes

3. Has the statistical analysis been performed appropriately and rigorously? 

Reviewer #3: Yes

Reviewer #4: Yes

4. Have the authors made all data underlying the findings in their manuscript fully available?

Reviewer #3: Yes

Reviewer #4: Yes

5. Is the manuscript presented in an intelligible fashion and written in standard English?

Reviewer #3: No

Reviewer #4: Yes

6. Review Comments to the Author

Reviewer #3: The Authors extensively answered on the Reviewers' comments which were very constructive. Only some minor typographical errors in the text and in the Table 1 need correction.

Reviewer #4: The present revised manuscript describes the results of a study investigasting the alteration of different immune marker in salvia with specific focus to SLPI and MMP-9 under different exercise conditions and hydration status. Beside oft hem the alteration of immune marker in the blood are shown. The study seems in principal well done and presented. The discussion and conclusion are a bit speculative. The relevance of the findings for exercise immunology isn’t fully clear. Nevertheless the findings are interesting and give new information about the alteration of specific immune marker in salvia. Limitations are given in the specific comments.

Specific comments:

1. From the present data it is difficult to conclude that SLPI amd MMP-9 could be a general marker for immunersponse or it is onlya marker for mucosa immune response. This should be pointed out.

2. It would be helpful to show direct correlations of the salvia and serum level.

3. In the figures T2, T3 are better given. Signifcant values should be labeled in the figures.

7. PLOS authors have the option to publish the peer review history of their article (what does this mean?). If published, this will include your full peer review and any attached files.

Reviewer #3: No

Reviewer #4: No

---

## [Author Response · Author response to Decision Letter 1]

19 Jul 2023

Review Comments to the Author

Reviewer #3: The Authors extensively answered on the Reviewers' comments which were very constructive. Only some minor typographical errors in the text and in the Table 1 need correction. The authors would like to thank again the reviewer for his thorough review which has improved the quality of the manuscript significantly. We have reviewed the paper for typo’s and hope to have all captured

Reviewer #4: The present revised manuscript describes the results of a study investigating the alteration of different immune marker in salvia with specific focus to SLPI and MMP-9 under different exercise conditions and hydration status. Beside oft hem the alteration of immune marker in the blood are shown. The study seems in principal well done and presented. The discussion and conclusion are a bit speculative. The relevance of the findings for exercise immunology isn’t fully clear. Nevertheless the findings are interesting and give new information about the alteration of specific immune marker in salvia. Limitations are given in the specific comments. The authors would like to thank again the reviewer for his thorough review which has improved the quality of the manuscript significantly

Specific comments:

1. From the present data it is difficult to conclude that SLPI and MMP-9 could be a general marker for immune response or it is only a marker for mucosa immune response. This should be pointed out. The authors agree that these markers are not general/mucosal immune markers, since they both have anti-inflammatory actions, in the abstract & overall conclusion ‘immune responses’ has been replaced with ‘(local) inflammatory responses’.

2. It would be helpful to show direct correlations of the salvia and serum level. We have studied the direct correlations (per timepoint) of the markers. The difficulty in this lies in the enormous difference in concentration between serum and saliva (which is why the graphs are presented as log-transformed), especially for SLPI which is extremely low in serum. Also, MMP-9 levels in saliva do not directly reflect blood concentrations since the pattern of increase is different in saliva from that in plasma (this sentence is mentioned in the discussion). For these reasons we decided that it did not contribute to the message of the paper and left it out. However, all data will be available for everyone who would like to perform other calculations with the data.

3. In the figures T2, T3 are better given. Significant values should be labelled in the figures. Significance (*) has been added to the figures

---

## [Decision Letter · Decision Letter 2]

29 Aug 2023

Salivary concentrations of Secretory Leukocyte Protease Inhibitor and Matrix Metallopeptidase-9 following a single bout of exercise are associated with intensity and hydration status

PONE-D-22-17078R2

Dear Dr. Knipping,

We’re pleased to inform you that your manuscript has been judged scientifically suitable for publication and will be formally accepted for publication once it meets all outstanding technical requirements.

Kind regards,

David M. Ojcius

Academic Editor

PLOS ONE

Additional Editor Comments (optional):

Reviewers' comments:

Reviewer's Responses to Questions

**Comments to the Author**

1. If the authors have adequately addressed your comments raised in a previous round of review and you feel that this manuscript is now acceptable for publication, you may indicate that here to bypass the “Comments to the Author” section, enter your conflict of interest statement in the “Confidential to Editor” section, and submit your "Accept" recommendation.

Reviewer #3: (No Response)

Reviewer #4: All comments have been addressed

2. Is the manuscript technically sound, and do the data support the conclusions?

Reviewer #3: (No Response)

Reviewer #4: Yes

3. Has the statistical analysis been performed appropriately and rigorously? 

Reviewer #3: (No Response)

Reviewer #4: Yes

4. Have the authors made all data underlying the findings in their manuscript fully available?

Reviewer #3: (No Response)

Reviewer #4: Yes

5. Is the manuscript presented in an intelligible fashion and written in standard English?

Reviewer #3: (No Response)

Reviewer #4: Yes

6. Review Comments to the Author

Reviewer #3: The Authors answered on the Reviewers' comments and improved the manuscript significantly. I have no further comments related to this manuscript.

Reviewer #4: no further comments. The revision of the revised manuscript is well done. Now the relvance of the findings are better given.

7. PLOS authors have the option to publish the peer review history of their article (what does this mean?). If published, this will include your full peer review and any attached files.

Reviewer #3: No

Reviewer #4: No

---

## [Editor Report · Acceptance letter]

27 Oct 2023

PONE-D-22-17078R2 

Salivary concentrations of Secretory Leukocyte Protease Inhibitor and Matrix Metallopeptidase-9 following a single bout of exercise are associated with intensity and hydration status 

Dear Dr. Knipping:

I'm pleased to inform you that your manuscript has been deemed suitable for publication in PLOS ONE. Congratulations! Your manuscript is now with our production department. 

Kind regards, 

on behalf of

Dr. David M. Ojcius 

Academic Editor

PLOS ONE